# Influence of HPT and Accumulative High-Pressure Torsion on the Structure and Hv of a Zirconium Alloy

Dmitry Gunderov [1], Andrey Stotskiy [2,*] , Yuri Lebedev [1] and Veta Mukaeva [2]

1 Institute of Molecule and Crystal Physics, Ufa Federal Research Center of the Russian Academy of Sciences, 450075 Ufa, Russia; dimagun@mail.ru (D.G.); lebedev@anrb.ru (Y.L.)
2 Ufa State Aviation Technical University, 450008 Ufa, Russia; veta_mr@mail.ru
* Correspondence: stockii_andrei@mail.ru

**Abstract:** The authors previously used the accumulative high-pressure torsion (ACC HPT) method for the first time on steel 316, β-Ti alloy, and bulk metallic glass vit105. On low-alloyed alloys, in particular, the zirconium alloy Zr-1%Nb, the new method was not used. This alloy has a tendency to $\alpha \rightarrow \omega$ phase transformations at using simple HPT. When using ACC HPT, the $\alpha \rightarrow \omega$ transformation can be influenced to a greater extent. This article studies the sliding effect and accumulation of shear strain in Zr-1%Nb alloy at various stages of high-pressure torsion (HPT). The degree of shear deformation at different stages of HPT was estimated. The influence of various high-pressure torsion conditions on the micro-hardness and phase composition by X-ray diffraction (XRD) of Zr-1%Nb was analyzed. It is shown that at high-pressure torsion revolutions of $n = 2$, anvils and the specimen significantly slip, which is a result of material strengthening. It was found that despite sliding, regular high-pressure torsion resulted in the high strengthening of Zr-1%Nb alloy (micro-hardness more than doubled), and after high-pressure torsion $n = 10$, up to 97% of the high-pressure ω-phase was formed in it (as in papers of other researchers). Accumulative high-pressure torsion deformation leads to the strongest transformation of the Zr-1%Nb structure and Hv and, therefore, to a higher real strain of the material due to composition by upsetting and torsion in strain cycles.

**Keywords:** Zr-1%Nb zirconium alloy; HPT; accumulative strain; mechanical properties



## 1. Introduction

The search for new materials or the improvement of already known materials for medicine is an increasing trend today. In addition to biocompatibility, the main requirements for medical implants are, firstly, a small discrepancy between the fixator/implant and the bone (lower modulus of elasticity in the area of contact of the material with the bone) and, secondly, this rather high strength for an internal fixation in case of bone fractures [1,2].

Severe plastic deformation (SPD) is a widely known method for improving mechanical properties. In addition, as is known [3,4], there are several SPD methods—high-pressure torsion, equal-channel angular pressing, multiaxial forging, differential velocity sideways extrusion, etc. They allow increasing the mechanical properties of many metals and alloys to the required level, which is undoubtedly an alternative to the search for new materials and their subsequent approbation.

High-pressure torsion deformation is a common method of structure transformation, nano-structural condition formation, and an increase in mechanical properties of metallic materials [3,5]. The HPT method is a laboratory method but it helps to evaluate materials in terms of maximum strengthening at high strains and their possible influence upon phase conversions [3,5]. When using the HPT method, strain parameters may vary, for example, applied pressure, the number of revolutions, strain temperature, anvil rotation speed, anvil geometry, the initial state of the strained material, the content of impurities in it [3,5], so the results for HPT of the same materials may be highly different in various authors at the

same pressure and number of revolutions. Zr and its alloys are actively used in nuclear power [6] and as bio-compatible materials for implants [7].

It is known that when the α-phase Zr is subjected to hydrostatic pressure, it transforms into a simple hexagonal structure (ω) at applied pressures ranging from 2 to 6 GPa and, finally, to the bcc β phase when the applied pressure exceeds 30 GPa [8]. Previously, the HPT effect on Zr and its alloys was researched, in particular in [9–15]. It was found that HPT of Zr and its alloys with Nb led to grain refinement to 30 nm and to α → ω phase transition. In this case, the pressure, at which α → ω occurs during HPT, is much lower than transition pressure without strain. Hence, strain activates α → ω transition in Zr. Moreover, the ω-phase is stabilized in the refined and HPT-hardened structure (reverse ω → α transition does not occur after pressure is relieved).

As the studies show [16–19], the actually achieved strain γ during HPT of hard or hardened metals and alloys can be much lower than as expected, which is predicted by the formula (1):

$$\gamma = 2\pi nR/h, \tag{1}$$

where $n$ is the number of revolutions, $R$ is the radius from the center to the measurement point, $h$ is the specimen thickness. Such inconsistency can be explained by the sliding effect of anvils over the specimen surface during HPT of hard and strain-hardened materials [16]. Furthermore, a new method was proposed in [20]—accumulative high-pressure torsion (ACC HPT) deformation to achieve high strains in hard and hardened materials. ACC HPT was applied and proved its efficiency on bulk metallic glass [20,21], and there is one paper where ACC HPT is applied to crystalline metallic materials [22], but it has not been applied to Zr and its alloys.

Further to this, a new method was proposed in [20]—accumulative high-pressure torsion (ACC HPT) deformation to achieve high strains in hard and hardened materials. This paper intended to determine, using a method from [19], the degree of shear strain achieved in Zr-1%Nb alloy during HPT, to study the structure and micro-hardness of the zirconium alloy subject to ACC HPT, and to compare ACC HPT and HPT effects in various conditions.

## 2. Materials and Methods

This research used E110 (Zr-1%Nb) low-alloyed zirconium alloy. E110 (Zr-1%Nb) zirconium alloy is a famous Russian biocompatible material. The chemical composition of this alloy is given in Table 1.

**Table 1.** Chemical composition of Zr-1Nb alloy (weight %).

| Zr | Nb | O | Hf | Fe | Ca | C | Ni | Cr | Si |
|------|------|------|------|------|------|------|------|------|------|
| Base | 1.1 | 0.1 | 0.05 | 0.05 | 0.03 | 0.02 | 0.02 | 0.02 | 0.02 |

Flat anvils (no groove, 20 mm in diameter) were used for HPT. HPT was done at room temperature at 2 and 6 GPa. HPT conditions are pressure and the number of revolutions n; specimens are designated in Table 2. Figure 1 shows a sketch of the HPT accumulative strain process.

**Table 2.** Strain conditions.

| Conditions/Designations | Method | Applied Pressure, GPa | Number of Revolutions |
|:---:|:---:|:---:|:---:|
| R1 | HPT | 2 | 10 |
| R2 | ACC HPT | 2 | $n\Sigma = 10$ |
| R3 | HPT | 6 | 1 |
| R4 | HPT | 6 | 10 |
| R5 | ACC HPT | 6 | $n\Sigma = 10$ |

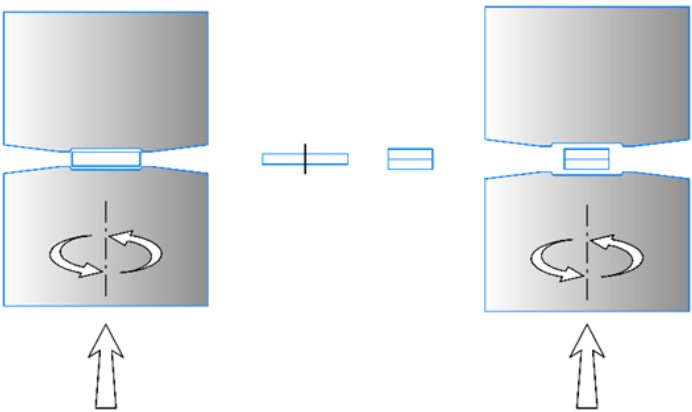

**Figure 1.** Sketch of the accumulative high-pressure torsion process.

The following procedure was used for ACC HPT straining (Figure 1): 1 HPT revolution → cutting the resulted disk specimen into parts → upsetting of stacked parts on anvils → HPT for 1 revolution. As a result, the material obtains a total strain by upsetting and torsion since the specimen invariably obtains high strain during upsetting and at the initial stage of torsion. Such cycles for specimens R2, R5 were undertaken five times. At the last stage, stacked segments were subject to simple HPT with $n = 5$, which resulted in the full consolidation of parts into a one-piece disk, and the total number of anvil revolutions for specimens R2, R5 was $n\Sigma = 10$. ACC HPT $n\Sigma = 5$ was also undertaken at $P = 2$ GPa and $P = 6$ GPa.

Micro-hardness was measured using the Vickers method under load 1 N (100 g) for 10 s. X-ray diffraction (XRD) analysis was conducted using a Rigaku Ultima IV diffractometer. The samples in the middle of the radius were examined with CuK$\alpha$-radiation (40 kV, 30 mA) and the phase composition of the alloy was determined using the Rietveld method.

## 3. Results and Discussion

As delivered, Zr-1%Nb alloy has a polycrystalline structure, consisting of the primary $\alpha$-Zr phase. Nb in the form of particles in the grains and at the grain boundaries may be present in the microstructure.

To find the actually achieved degree of strain by torsion, two halves of the alloy disk (Figure 2a) were subject to combined HPT by 90° ($n = 1/4$) according to the scheme (Figure 2b) [19]. After combined HPT by $n = 1/4$ revolutions, visual inspection of the specimen (according to the relative shear of the upper and lower surfaces of the halves) shows that the specimen has received the degree of shear strain $\gamma = 3$, which is 2.6 times lower than $\gamma = 7.8$ as predicted by formula (1) ($\gamma$ is evaluated in point $R = 5$, $h = 1$ mm). A lower strain is caused by the sliding effect. In the second experiment, the alloy disk was at first subject to HPT 2 revolutions, then the specimen was cut and the resulting halves were subject to combined HPT $n = 1$ revolution. The specimen was warped, but the expected shear of the surfaces of halves did not occur. According to the relative shear of the upper and lower surfaces of the halves, it can be evaluated that the specimen has received the degree of shear strain $\gamma = 4$, which is 20 times lower than $\gamma = 80$ as predicted by formula (1) ($\gamma$ is evaluated in point $R = 1$, $h = 1$ mm). In the third experiment, the alloy disk was at first subject to HPT 5 revolutions, then the specimen was cut and the resulting halves were subject to combined HPT $n = 1/4$ revolution. The specimen was warped, but the expected shear of the surfaces of halves did not occur, since the shear strain $\gamma$ expected by formula (1) was not realized during HPT.

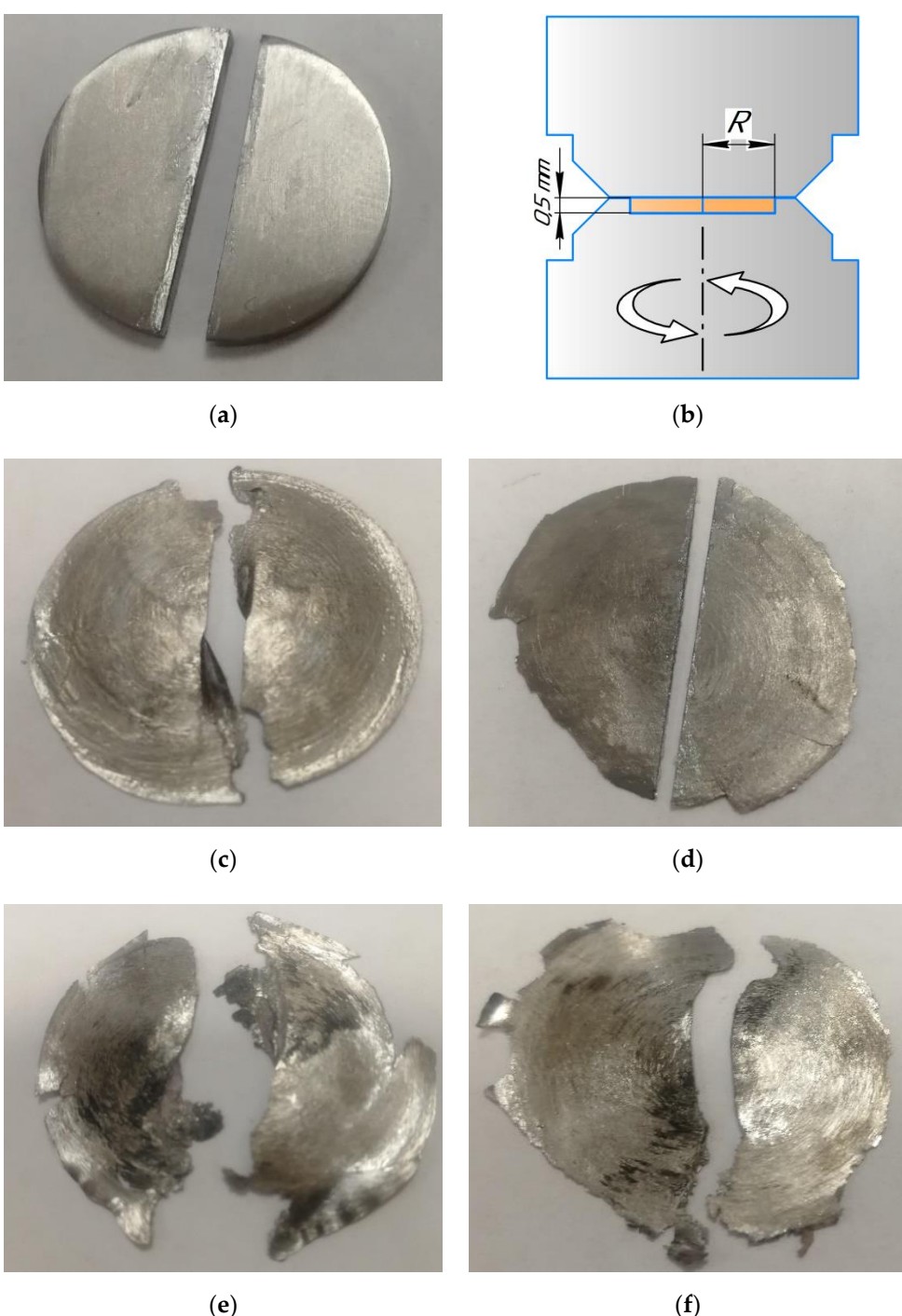

**Figure 2.** (**a**) two halves of the Zr-1%Nb disk for combined HPT, (**b**) sketch of combined HPT, (**c**) two halves after combined HPT $n = 1/4$, (**d**) two halves after specimen cutting, HPT $n = 2$; (**e**) halves after HPT $n = 2$ + cutting + combined HPT $n = 1$; (**f**) halves after HPT $n = 5$ + cutting + combined HPT $n = 1/4$.

The observed effect can be explained as follows: specimen shear strain during HPT is done if the friction force $F_\mu$ between the anvil surface and specimen will exceed the specimen material yield stress ($\sigma$). The friction force $F_\mu$ between the anvil surface and specimen is found by equation $F_\mu = P\mu$, where $P$ is pressure, $\mu$ is the friction coefficient. The pressure was evaluated as $P = U/S$, where $U$ is the press force (200 t), $S$ is the anvil force ($d = 2$ cm, $S = 2.4$ cm$^2$). Then $P$ is 6 GPa. However, in reality, the zone up to $d = 3$ cm, $S = 7.2$ cm$^2$ fits into the pressure effect zone taking into account edges of the anvil working

area and material dross. Taking this into account, *P* is 2.7 GPa. The friction coefficient for metallic materials can be taken as $\mu$ = 0.4. Hence, in this case, the friction force between the anvil surface and specimen is $F_\mu$ = 1.1 GPa. This is higher than the yield stress of initial Zr-1%Nb (0.3 GPa). As the number of HPT revolutions rises, the material becomes more and more hardened. After HPT *n* = 2, microhardness (Hv) reaches about 3–3.5 GPa, so the yield stress (found under the known formula $\sigma$ = Hv/3) $\sigma$ = 1 GPa. After HPT *n* = 5, microhardness (Hv) reaches about 3.5–4 GPa, so the yield stress $\sigma$ = 1.1–1.3 GPa, which is higher than the friction force. This causes an increased sliding effect as HPT grows, and for HPT *n* = 5, shear strain is not realized. The result supports the relevance of using accumulative HPT.

After HPT for all conditions, micro-hardness rises in all edge areas of HPT disks by more than twice, which proves strong refinement of the structure (Table 3). For R1 and R3 conditions, lower Hv values are observed in the center of specimens as compared to edge areas. In the case of R1 conditions, it seems to be insufficient for homogeneous refinement of the specimen structure in diameter due to insufficient pressure. In the case of R3, only one revolution was done at 6 GPa, which is insufficient to refine the material in the central area. For R2, R4, R5 conditions, Hv values in specimen diameters become closer, e.g., specimens become more homogeneously strained along the entire radius. The highest growth of Hv occurs after accumulative HPT *n* = 10 at 6 GPa (R5), which proves a higher strain introduced into the material, manly in comparison with HPT *n* = 10, *P* = 6 GPa (R4). Hence, accumulative strain results in a higher actual strain and additional refinement of the structure due to composition by upsetting and torsion in strain cycles.

**Table 3.** Results of the XRD analysis and microhardness of Zr-1Nb alloy in different conditions.

| Treatment | X-ray Analysis | | Microhardness, HV | | |
|---|---|---|---|---|---|
| | $\alpha$-Phase, % | $\omega$-Phase, % | Center | 0.5R | Edge |
| Initial state | 100 | - | | 149 $\pm$ 5 | |
| R1 (2 GPa, *n* = 10) | 20 | 80 | 283 $\pm$ 35 | 366 $\pm$ 17 | 343 $\pm$ 13 |
| R2 ACC (2 GPa, *n* = 10) | 18 | 82 | 347 $\pm$ 14 | 362 $\pm$ 24 | 346 $\pm$ 22 |
| R3 (6 GPa, *n* = 1) | 7 | 93 | 300 $\pm$ 7 | 375 $\pm$ 13 | 372 $\pm$ 21 |
| R4 (6 GPa, *n* = 10) | 4 | 96 | 371 $\pm$ 15 | 372 $\pm$ 6 | 368 $\pm$ 15 |
| R5 ACC (6 GPa, *n* = 10) | 3 | 97 | 410 $\pm$ 17 | 385 $\pm$ 19 | 396 $\pm$ 24 |

As said above, according to XRD, Zr-1%Nb alloy contained the $\alpha$-phase (Table 3), also, a very small amount of hydride phases based on zirconium may appear in the material [23], as shown by the appearance of a small peak between 33–35°, their calculation has not been performed. After HPT in all conditions, the $\omega$-phase appears in the amount of 80% and more due to $\alpha \rightarrow \omega$ transition (Figure 3), as earlier shown in [9–13]. The studies [10] show that $\alpha \rightarrow \omega$ transition is possible for pure Zr under the action of HPT at room temperature under 6 GPa, then no $\omega$-phase occurred during HPT at 2 GPa. In the experiment, the $\omega$-phase occurred as early as at HPT *P* = 2 GPa (Table 3). Hence, the difference of 1% Nb in Zr leads to $\alpha \rightarrow \omega$ phase transition at lower HPT pressure (*P* = 2) than in the case of HPT of pure Zr [10]. In the Zr-1Nb structure after SPD, the appearance of a $\beta$-phase is also possible, however, due to the superposition of the peaks of the $\beta$ and $\omega$ phases, its identification is an ambiguous task. By comparing states R1 and R4, a conclusion can be made that higher HPT pressure (6 GPa) leads to an increased fraction of the $\omega$-phase (up to 96%). A similar influence of pressure was found earlier [10]. It should be also noted that HPT *P* = 6 GPa, *n* = 1 (R3) leads to the formation of a higher fraction of the $\omega$-phase (93%) than HPT *P* = 2 GPa, *n* = 10 (R1) or ACC HPT *P* = 2 GPa (R2), which shows a high role of the strain degree (n) in $\omega$-phase formation and applied pressure in these HPT conditions. Let us note that the content of the $\alpha$-phase in samples R4 (HPT *P* = 6 GPa, *n* = 10) and R5 (ACC HPT *P* = 6 GPa, *n* = 10) becomes minimal, which is due to the highest degree of deformation during HPT and more intense phase transformations of the structure in comparison with other regimes.

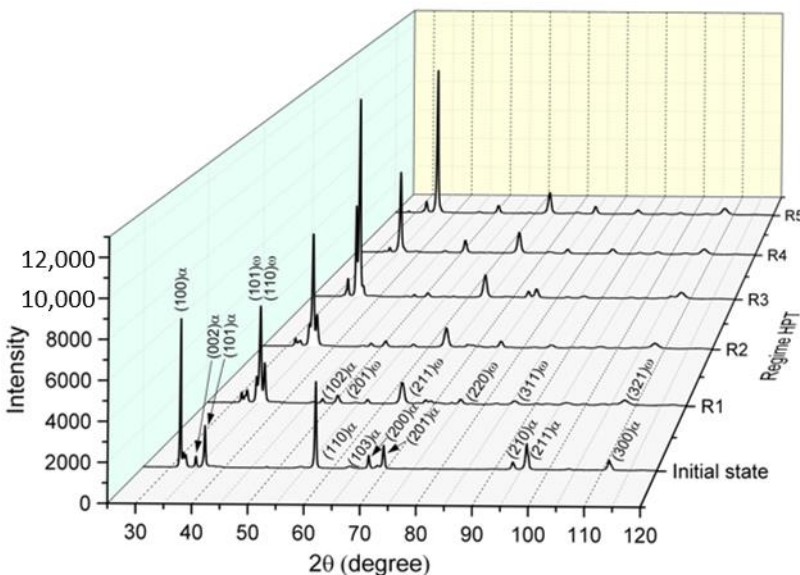

**Figure 3.** X-ray diffractograms of Zr-1Nb alloy in the initial state and in various HPT conditions.

### 4. Conclusions

Thus, the following can be noted: it was shown that high sliding starts and very low shear strain is realized during HPT of Zr-1%Nb alloy after $n = 2$ revolutions of anvils and the shear strain is not realized after HPT $n = 5$. At the same time, for HPT $P = 6$ GPa, $n = 10$ of Zr-1%Nb alloy, the structure and properties are changed (increased Hv > 370 MPa, the fraction of the $\omega$-phase > 96%) similar data are observed in Zr-Nb alloys during HPT in the same conditions in other papers [12–14]. Hence, despite sliding at $n = 5$, strain occurs during HPT. It can be suggested that in other papers, during HPT of similar alloys and metals after reaching some value by the number of anvil revolutions, sliding starts and shear strain is not realized, but the sliding effect was not evaluated. It seems that strain during HPT in these cases took place by another method, not by shear according to formula (1). The method [19] allows a simple evaluation of the sliding effect during HPT at various stages.

For the Zr-1% Nb alloy, the accumulative high-pressure torsion method was applied for the first time. According to XRD data and Hv, accumulative HPT leads to the strongest transformation of the Zr-1%Nb structure and Hv and, therefore, to a higher real strain of the material due to combination by upsetting and torsion in strain cycles.

**Author Contributions:** Conceptualization, D.G.; methodology, D.G.; validation, D.G., A.S. and Y.L.; formal analysis, D.G., A.S. and Y.L.; investigation, D.G., A.S. and Y.L.; resources, V.M.; data curation, D.G.; writing—original draft preparation, D.G., A.S. and V.M.; visualization, A.S.; supervision, D.G.; project administration, V.M.; funding acquisition, V.M. All authors have read and agreed to the published version of the manuscript.

**Funding:** This research was funded by the Russian Science Foundation, No 20-79-10189.

**Institutional Review Board Statement:** Not applicable.

**Acknowledgments:** Authors are grateful to the personnel of the research and technology Joint Research Center, 'Nanotech', Ufa State Aviation Technical University for their assistance with instrumental analysis.

**Conflicts of Interest:** The authors declare no conflict of interest.

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
