# Peer review of "Influence of HPT and Accumulative High-Pressure Torsion on the Structure and Hv of a Zirconium Alloy"

_metals, doi:10.3390/met11040573_

Round 1

Reviewer 1 Report

Dear Editor,

Thе authors of the manuscript compered the sliding effect and accumulation of shear stress in Zr-1Nb alloy subjected to high pressure torsion deformation and accumulative high pressure torsion in various conditions. The manuscript contains the original and interesting results. My recommendation is minor revision. My questions and recommendations are presented below.

1) The chemical composition of the alloy shown in Table 1, is weight percentages or atomic? How was the chemical composition of the alloy measured? What is the measurement error?

2) Please correct the number of the table in the sentence “specimens are designated in Table 1.” (line 65). Table 1 presents only the chemical composition of the alloy.  

3) Table 3 shows the erroneous designation of the volume fraction of the beta and omega phases. Please correct it.

4) Ref. 19 is not used in the text of the manuscript.

Author Response

The authors thank reviewer for the questions and comments.

Point 1: The chemical composition of the alloy shown in Table 1, is weight percentages or atomic? How was the chemical composition of the alloy measured? What is the measurement error?

Response 1: According to the manufacturer's certificate the chemical composition in Table 1 is measured in weight percent. The name of the table will be indicated (weight%).

Point 2: Please correct the number of the table in the sentence “specimens are designated in Table 1.” (line 65). Table 1 presents only the chemical composition of the alloy.  

Response 2: The number of the table in the sentence “specimens are designated in Table 2.” has been corrected.

Point 3: Table 3 shows the erroneous designation of the volume fraction of the beta and omega phases. Please correct it.

Response 3: The designations for the volume fraction of phases have been corrected in table 3.

Point 4: Ref. 19 is not used in the text of the manuscript.

Response 4: Ref. 19 was not indicated as a reference in the text. We corrected it.

Reviewer 2 Report

The abstract: it is not clear why this research is conducted after reading the abstract. Brief background and research methods should be introduced first.

Introduction: The literature review could be extended, not just limiting to HPT. I suggest add more relevant literature review, covering studies on SPD (severe plastic deformation). Regarding grain refinement, I suggest consider the article: doi: 10.1016/j.ijmachtools.2019.03.002.

Materials and Methods:

What is ACD? More details about the experimental methods and procedures should be given.

The principle of AHPT should be illustrated schematically.

The locations for Micro-hardness should be illustrated.

The locations for XRD should be given.

Author Response

The authors thank reviewer for the questions and comments.

Point 1: The abstract: it is not clear why this research is conducted after reading the abstract. Brief background and research methods should be introduced first.

Response 1: The abstract has been revised for a clearer understanding of this study.

Point 2: Introduction: The literature review could be extended, not just limiting to HPT. I suggest add more relevant literature review, covering studies on SPD (severe plastic deformation). Regarding grain refinement, I suggest consider the article: doi: 10.1016/j.ijmachtools.2019.03.002.

Response 2: It is more logical to compare the DVSE method with the ECAP method, since both deformation methods are similar. The DVSE method is more suitable for metals and alloys with low strength, as shown in the article (doi: 10.1016 / j.ijmachtools.2019.03.002) for the aluminum alloy AA1050. HPT is a more specific deformation method in which significantly higher deformation is achieved due to high pressure and torsion at the same time, therefore, this method can be used to deform strong and high strength alloys.

Point 3: What is ACD? More details about the experimental methods and procedures should be given.

Response 3: ACD has been replaced by AHPT in the text and table 2.

Point 4: The principle of AHPT should be illustrated schematically.

Response 4: The principle of AHPT was illustrated schematically in figure 1. The description of this deformation method is presented in detail in the “materials and methods”.

Point 5: The locations for Micro-hardness should be illustrated.

Response 5: The locations of microhardness measurements were indicated in Table 3.

Point 6: The locations for XRD should be given.

Response 6: We have added the location of XRD to the materials and methods.

Reviewer 3 Report

The  acumulative high-pressure torsion (AHPT) technique that is presented in this manuscrit is very interesting for basic research. Part of the examination of the formula for calculating the deformation for HPT is also interesting, but I think that the conclusions need to be formulated carefully, unless this is examined in detail by the wider scientific community.

I have only two comments on the work:

In Table 3, all phases are marked as alpha, the designation of beta and omega phases is missing.

The chapter "conclusion" is missing, I require it to be added to the manuscript.

Author Response

The authors thank reviewer for the questions and comments.

Point 1: In Table 3, all phases are marked as alpha, the designation of beta and omega phases is missing.

Response 1: An error was made in Table 3, the phase designations were corrected.

Point 2: The chapter "conclusion" is missing, I require it to be added to the manuscript.

Response 2: We've added a conclusion chapter.

Reviewer 4 Report

The manuscript is on the effect of HPT and AHPT on micro-hardness and phase composition/transformations of a Zr-1%Nb alloy. X-ray data and micro-hardness measurements show that the beta phase in this alloy transforms into the alpha and omega phases upon HPT. Mechanical behavior was explored using micro-hardness. Phase transformations are also taken into account employing X-ray measurements. Experimental method is well described and obtained results were sufficiently discussed. References are updated and sufficiently exhaustive. However, no microstructural studies were conducted in this research work. I suggest the authors add a few micrographs showing the microstructure of this alloy before and after HPT and AHPT. The questions listed below should be addressed before publication:

Is there any grain refinement due to the HPT?

Does the grain size vary from center to edge upon HPT?

Is the level of grain refinement similar in HPT and AHPT? If not, why?

A “Conclusions” section should also be added into the paper.

Author Response

The authors thank reviewer for the questions and comments.

Point 1: Is there any grain refinement due to the HPT?

Does the grain size vary from center to edge upon HPT?

Is the level of grain refinement similar in HPT and AHPT? If not, why?

Response 1: Our further research will focus on a detailed study of the microstructure after SPD and its features. This article aims more at SPD evaluating, including AHPT Zr-1Nb.

Undoubtedly, the HPT method leads to a strong refinement of the Zr-1Nb structure, which is indirectly indicated by the measurement of the microhardness, which more than doubles. It is also known that at a low number of HPT revolutions, a weaker refinement of the structure occurs in the center; at a high degree of deformation, this effect is leveled. Exactly the next direction of our research will be to study the influence of HPT and especially AHPT on microstructure refinement.

Point 2: A “Conclusions” section should also be added into the paper.

Response 2: The "Conclusions" section has been added to the article.

Round 2

Reviewer 2 Report

The authors have improved the article. However, some comments still went unheeded:

The abstract: it is not clear why this research is conducted or what is the application/significance after reading the abstract. Brief background should be introduced first.

The authors were invited to briefly add more introduction to the available SPD methods (such as ECAP, DVSE etc) to increase the relevance of the research topic and attract more readers. It is not necessary to compare different SPD methods, but a brief introduction is necessary. As such, I suggest add one short paragraph in the very beginning introducing the background of SPD, why is SPD being studied and what is the application/significance.

Fig 1 is captioned as HPT but not AHPT.

Author Response

Point 1. The abstract: it is not clear why this research is conducted or what is the application/significance after reading the abstract. Brief background should be introduced first.

Response 1. The authors finalized the abstract, a brief background was added.

Point 2. The authors were invited to briefly add more introduction to the available SPD methods (such as ECAP, DVSE etc) to increase the relevance of the research topic and attract more readers. It is not necessary to compare different SPD methods, but a brief introduction is necessary. As such, I suggest add one short paragraph in the very beginning introducing the background of SPD, why is SPD being studied and what is the application/significance.

Response 2. The introduction has been revised. Paragraphs on biomaterials and SPD methods were added.

Point 3. Fig 1 is captioned as HPT but not AHPT.

Response 3. Fig.1 was changed: Sketch of the accumulative high-pressure torsion process

Reviewer 3 Report

The author accepted my comments from the revision. 

Author Response

The authors are grateful to the reviewer for comments on the article.

Reviewer 4 Report

Since the authors will focus on a follow-up study for microstructural evolution during HPT, the present work can be accepted.

Author Response

The authors are grateful to the reviewer for comments on the article. Our next research will be focused on the microstructure investigation.

Round 3

Reviewer 2 Report

Thank you for the response. I have no further comments.